# Expression of Chitinase and shRNA Gene Exhibits Resistance to Fungi and Virus

**DOI:** 10.3390/genes14051090

**Published:** 2023-05-15

**Authors:** Samia Parveen, Anwar Khan, Nusrat Jahan, Khadija Aaliya, Adnan Muzaffar, Bushra Tabassum, Syed Inayatullah, Syed Moeezullah, Muhammad Tariq, Zainia Rehmat, Niaz Ali, Abrar Hussain

**Affiliations:** 1Department of Biotechnology, Balochistan University of Information Technology, Engineering and Management Sciences, Quetta 87300, Pakistan; 2Department of Microbiology, Balochistan University of Information Technology, Engineering and Management Sciences, Quetta 87300, Pakistan; 3Center of Excellence in Molecular Biology, University of the Punjab, Lahore 54000, Pakistan; 4School of Biological Sciences, University of the Punjab, Lahore 54000, Pakistan; 5Department of Biotechnology, Sardar Bahadur Khan Women’s University Balochistan, Quetta 87300, Pakistan; 6Department of Botany, Hazara University, Mansehra 21300, Pakistan

**Keywords:** *Fusarium oxysporum*, shRNA, virus

## Abstract

With the increasing global population, saving crops from diseases caused by different kinds of bacteria, fungi, viruses, and nematodes is essential. Potato is affected by various diseases, destroying many crops in the field and storage. In this study, we developed potato lines resistant to fungi and viruses, Potato Virus X (PVX) and Potato Virus Y (PVY), by inoculating chitinase for fungi and shRNA designed against the mRNA of the coat protein of PVX and PVY, respectively. The construct was developed using the pCAMBIA2301 vector and transformed into AGB-R (red skin) potato cultivar using *Agrobacterium tumefaciens*. The crude protein extract of the transgenic potato plant inhibited the growth of *Fusarium oxysporum* from ~13 to 63%. The detached leaf assay of the transgenic line (SP-21) showed decreased necrotic spots compared to the non-transgenic control when challenged with *Fusarium oxysporum*. The transgenic line, SP-21, showed maximum knockdown when challenged with PVX and PVY, i.e., 89 and 86%, while transgenic line SP-148 showed 68 and 70% knockdown in the PVX- and PVY-challenged conditions, respectively. It is concluded from this study that the developed transgenic potato cultivar AGB-R showed resistance against fungi and viruses (PVX and PVY).

## 1. Introduction

Potato, the world’s number one nongrain food, stands fourth after wheat, rice, and maize [1,2,3] as the most popular food worldwide, and is grown in more than 100 countries in Europe, Asia and elsewhere. As a rich carbohydrate source, it also provides vitamins such as B6, C, iron, carotenoids, phenolic acid, magnesium, potassium, and dietary fibres [4,5]. Due to its high protein and energy per unit area, potato is considered a valuable crop, ensuring safe food worldwide and gaining interest because it is comparably easy to grow and has a beneficial impact on the nutrition of humans [6], with more than 5000 varieties available around the world. In Pakistan, potatoes are grown in hilly to flat areas during the summer, autumn, and spring. It is an ideal crop because it requires little labour and can be harvested in less than 90 days after planting [7]. More than 40 diseases affect potato crops, with bacteria, fungi, viruses, and nematodes severely affecting all parts of Solanum tuberosum (potato) plants [8]. Disease-related losses are estimated to reach 50% in developing countries and 25% in Western countries. The most dangerous diseases in potatoes are caused by fungi and viruses, which are resistant to treatment and persist in future generations of tubers [9]. In addition, several soil-borne fungi remain as spores or reproductive agents unless conditions are favourable. Phytophthora, Pythium, Verticillium, Rosellinia, Sclerotinia, Fusarium, and Rhizoctonia are among them. Almost 40 virus species have been reported to affect potato plants worldwide [10,11]. PVX (Potato virus X), PVA (Potato virus A), PVS (Potato virus S), PVY (Potato virus Y), PVM (Potato virus M), PLRV (Potato aucuba mosaic virus), TRV (Tobacco rattle virus), TNV (Tobacco necrosis virus), and PMTV are some of them (Potato mop-top virus) [2].

A single PVX virus infection can result in a 10–40% loss in the potato crop, but when combined with PVY [3], it can result in a 55% reduction in yield [12]. PVY (potato virus Y) is transmitted in potatoes by winged aphids or mechanical means [5], and the virus remains infective for a few minutes or hours [13], causing 10–80% loss [12]. Because of the importance of potatoes as a food commodity, disease control is critical. To effectively control diseases, it is also necessary to obtain a thorough understanding of pathogens [8]. An efficient way to control viruses in potatoes is to use virus-free tubers [2,13].

Some of the practices for disease control in potatoes, such as using chemicals, biological control, and introducing resistant varieties, are critical to increasing crop yield [14]. In addition, in the modern era, genome modification is used to address global changes related to population growth by utilizing green revolution technology [15,16,17].

The physiological, cytological, and molecular studies of host–pathogenic interactions provided the impetus for developing disease-resistant crop varieties. Five pathogen-related (PR) subclasses (PR1-PR5) are essential in plant defence, with various functions. Among these, two essential proteins, chitinases (PR-3) and 1,3 glucanases (PR-2), catalyse the hydrolysis of essential polysaccharides of the cell wall, known as chitin and glucans [18].

RNA interference is a powerful anti-viral multiplication defence. Two RNA-mediated gene silencing procedures, RNA-directed DNA methylation and RNA-directed RNA degradation, provide highly efficient and high-yielding technology for gene suppression in plants [19].

To test resistance against the conserved coat protein gene of potato virus Y, the short hairpin RNA (shRNA) was expressed with CaMV35S promotor into a binary vector with a NOS terminator [4,20]. In vivo shRNA contains stem and loop sequences from a microRNA (miR304) for stability and shape. Cardinal, a potato variety transformed with the Agrobacterium construct, showed decreased mRNA expression for the PVY coat protein [5].

In this study, a transgenic potato cultivar AGB-R was developed using RNAi for potato virus X and Y and the PR gene, chitinase, for fungi in the plant binary vector pCAMBIA2301 via Agrobacterium-mediated transformation.

## 2. Materials and Methods

### 2.1. Construct Development

A construct containing the chitinase gene and shRNAs for PVX and PVY was constructed by combining the mentioned genes into pCAMBIA2301, already available in the Applied Biotechnology laboratory, BUITEMS, Quetta. Figure 1 shows the construct details. The construct after the ligation of the three genes was labelled as the pCAMBIA-CXY construct. The binary construct was propagated in *E. coli*, and positive colonies containing our genes of interest were confirmed by restriction digestion and further verified by sequencing. The *Agrobacterium tumefaciens* LB4404 was transformed with the pCXY construct via the freeze–thaw method [21]. The colonies harbouring pCAMBIA-CXY were screened using primers mentioned in Table 1.

### 2.2. Plant Transformation

The potato variety AGB-R was used to develop fungi- and virus-resistant potato plants. AGB-red plants were sub-cultured in vitro using plant nodal segments on MS medium supplemented with 3% sucrose and 2.44 g/L phytagel. The plants were grown in a 22 °C growth chamber with 16 h of light and 8 h of darkness. Every three weeks, the plants were sub-cultured. The potato variety AGB-R was transformed with a pCAMBIA-CXY construct using *Agrobacterium tumefaciens* LB4404 as described by [22]. Three hundred and twenty explants were transformed with *Agrobacterium tumefaciens* harbouring the pCAMBIA-CXY construct. In separate experiments, control potato plants of cultivar AGB-R were transformed with plasmid pCAMBIA2301 without the CXY construct. The transformed nodes were cultured and sub-cultured in MS medium after every three weeks and further shifted to pots in a greenhouse for hardening purposes.

### 2.3. Molecular Analysis of Transformed Plants

DNA was isolated according to [23] from the potentially transformed plants grown in the greenhouse, followed by PCR for the confirmation of transformed plants. The chitinase gene was amplified under the following cycle conditions: 95 °C for the first denaturation for 4 min, followed by 35 cycles each of denaturation at 95 °C for 30 s, annealing at 61 °C for 45 s and extension at 72 °C for 45 s. The final extension was performed at 72 °C for 7 min. The TBE buffer was used to resolve the amplified PCR product in 1% agarose gel, which was later observed under UV illumination to confirm the transformed gene.

The DIG-labelled probe was hybridized with 30 μg of genomic DNA isolated from PCR-positive, using the GenJET Plant Genomic DNA purification kit (cat#K0791; thermos scientific) and control plants to reveal the integration of the chitinase gene and shRNAs in transgenic potato lines. The genomic DNA was digested and resolved on agarose gel overnight with *EcoRI* and *HindIII* restriction endonuclease. The hybridization was performed for 16 h at 42 °C. 5-Bromo-4-chloro-3-indolyl phosphate/nitro blue tetrazolium (BCIP/NBT) was used to detect the hybridization signals.

### 2.4. Southern Blot

Southern hybridization was employed to evaluate the integration of the CXY construct within the genome of potato plants, as mentioned in [24]. To synthesize the digoxygenin (DIG)-labelled probe (Merck KGaA, Darmstadt, Germany), 672 bp of a chitinase sequence of the CXY construct was used. The hybridization was performed at 48 °C with denatured genomic DNA of transgenic and non-transgenic control plants. An alkaline phosphatase-conjugated antibody was used to detect the probe–target hybrid.

### 2.5. Bioassay for Chitinase Gene

#### 2.5.1. Fungal Inhibition Assay of Crude Plant Extract

The fungal (*Fusarium oxysporum*) inhibition of the crude protein of PCR-confirmed transgenic plants was performed according to [22]. Two to three leaves of PCR-positive transgenic plants grown in the greenhouse were ground in liquid nitrogen. Protein extraction was performed using a Thermo Scientific kit (Thermo Fisher Scientific, Waltham, MA, USA) as per the manufacturer’s protocol. Wells were formed in the centre of agar plates using a cork borer. A spore suspension of *F. oxysporum* was added into the well at 10^6^ spores/mL, and different volumes of transgenic crude protein were added ranging from 40 µL to 200 µL were added subsequently. The plates were incubated for five days at 28 °C, plates were observed, and the mycelium diameter was measured. Finally, the crude protein from non-transgenic plants was added as a negative control into the wells with the tested fungi. The calculation of percent inhibition was completed using the formula mentioned below [25].
$$Inhibition\;\%=\left\{100\times \left.\left(\frac{C-T}{C}\right)\right\}\right.$$

#### 2.5.2. Detached Leaf Assay

Fresh leaves of transgenic and non-transgenic control plants grown in greenhouses were detached and placed in sterile moist filter paper in Petri plates. The midribs of the leaves were injured and 20 μL of 106 spores/mL were inoculated. The Petri plates containing the inoculated leaves were placed under a 16/8 h light/dark cycle at 25 °C and 70% relative humidity. The severity of the disease was recorded 7 days post-inoculation.

#### 2.5.3. Endo Chitinase Assay

The endochitinase activity of transgenic potato plants was evaluated with the presence of recombinant chitinase protein. The endochitinase kit (Merck KGaA, Darmstadt, Germany) was used as per the manufacturer’s protocol. The autoclaved transgenic plant proteins were used as a negative control, while *Trichoderma* purified chitinase provided with the kit was used as a positive control.

### 2.6. Bioassays for Potato Viruses

The pathogen assay of transgenic potato plants cv. AGB-R was performed using potato virus X and Y pathogens. Dr Bushra Tabassum, School of Biological Sciences, University of the Punjab, Lahore, Pakistan, provided the infected plants.

#### 2.6.1. Infestation Assay of PVX

The infestation assay of PVX was performed according to [4]. The pathogen assay was conducted on 21 transgenic potato lines quarantined in the greenhouse. Three non-transgenic control plants were also infected simultaneously. The infected PVX leaves were ground in the presence of 1% phosphate buffer saline and the sap was used to infect the transgenic and non-transgenic control plants mechanically. Three lower leaves of the tested plants were inoculated. The infected potato plants were kept in the greenhouse at 22  ±  2 °C with a 16 h photoperiod. About three to four weeks post PVX inoculation, new leaves were used for ELISA (enzyme-linked immunosorbent assay).

#### 2.6.2. Infestation Assay of PVY

The infestation assay of PVY was performed according to [5]. The pathogen assay was conducted on 21 transgenic potato lines (cv. AGB-red) quarantined in the greenhouse. Three non-transgenic control plants were also infected simultaneously. Three lower leaves of the tested plants were inoculated with 5 μg/mL PVY with 995 μL of sodium-phosphate buffer (pH 7.4). The infected potato plants were kept in the greenhouse at 22 ± 2 °C with a 16 h photoperiod. About three to four weeks post PVY inoculation, new leaves were used for ELISA (enzyme-linked immunosorbent assay).

#### 2.6.3. Serological Assay

The total crude protein of infested transgenic and non-transgenic control plants was isolated from the young leaves of infested potato plants with PVX and PVY. The ELISA test was performed using anti-PVX polyclonal IgG and anti-PVY polyclonal IgG antibodies as per the manufacturer’s protocol (Bioreba, Basel, Switzerland). The absorbance was recorded at 405 nm after 40 min of substrate addition.

### 2.7. Expression Analysis Quantitative PCR

Thirty days post-infestation of PVX and PVY in transgenic and non-transgenic potato plants, qPCR was performed to evaluate the relative expression of the coat protein gene of PVX and PVY in virus-infested potato plants. RNA was isolated using Trizol reagent and 0.1 μg of RNA was used to synthesize cDNA using Revert Aid First Strand cDNA Synthesis Kit (Thermo Fisher Scientific, Waltham, MA, USA). The primers used for PVX and PVY qPCR are mentioned in Table 1. SYBR Green qPCR master mix (Thermo Fisher Scientific, Waltham, MA, USA) was used in the PikoReal Tm 3.1 software of the Thermo Scientific™ PikoReal™ Real-Time thermocycler. In addition, the housekeeping gene beta-actin was used as an internal control for normalization purposes. The mean and standard deviation were calculated using Ms. Excel and one-way ANOVA was applied through GraphPad Prism version 5.0 with the *p*-value < 0.05.

## 3. Results

This study aimed to develop transgenic potato lines tolerant to fungi and viruses (PVX and PVY). For fungi, the chitinase gene was used (codon optimized), while for PVX and PVY, shRNA containing sense and anit-sense strands within the coat protein gene of the respective virus was used.

### 3.1. Transformation of the pCAMBIA-CXY Construct

The ‘AGB-R’ potato variety was transformed with chitinase gene, shRNA PVX and shRNA PVY via Agrobacterium-mediated transformation. Figure 2 summarizes the regeneration, transformation, and development of potato plants. A total of 320 nodes from AGB-R were transformed by means of cocultivation with Agrobacterium tumefaciens LB4404. In total 162 plantlets regenerated into complete plants after transformation with the construct of interest. The transformation of pCAMBIA-CXY into AGB-R was confirmed by the amplification of the chitinase gene in the PCR reaction. Figure 3A shows the PCR amplification of the ~672 bp chitinase gene in the transformed potato plants. Among the 162 regenerated plants, only 30 plants were PCR-positive.

The stable integration of the construct in the potato variety AGB-R was detected by Southern blot. The gene integration was detected using a chitinase gene-specific probe. Upon using a DIG-labelled chitinase-specific probe, all of the PCR-positive plants showed stable integration, as shown in Figure 3B.

### 3.2. Fungal Bioassay

#### In Vitro Antifungal Activity of Transgenic Plants

The PCR-positive plants were acclimatized in pots and placed in a quarantine greenhouse. The temperature was maintained at 22 °C with 70% relative humidity, while the light and dark period was maintained at 16 h and 8 h, respectively. The crude protein from the transgenic and non-transgenic control plants was isolated and about 200 µg of crude protein was added to the agar wells with 106 spores/mL of *Fusarium oxysporum*. All of the transgenic plants inhibited the fungus, but the extent of inhibition varied among tested plants. Figure 4 shows the hyphal growth inhibition of *Fusarium oxysporum* among transgenic potato plants. The transgenic plant SP-21 showed the maximum inhibition, i.e., 62.8% of the radial growth of the fungus, while SP-89 showed the minimum inhibition, i.e., 13.65% of the radial growth of the fungus compared to the non-transgenic control.

An endochitinase activity assay was performed using a crude protein of the transgenic potato plants. The assay defines the chitinase enzyme present in the transgenic potato plant which hydrolyses the chitinase substrate p-nitrophenyl β-D-N,N′,N″-triacetylchitotriose. The maximum value of the chitinase activity was observed for the positive control provided in the kit, i.e., 6.28 U. The negative control was the substrate (p-nitrophenyl β-D-N,N′,N″-triacetylchitotriose)-only solution. The assay revealed that all of the PCR-positive transgenic plants contained endo-chitinase enzyme activity, with the maximum activity observed for SP-021 (1.19 U) and the minimum activity observed for SP-089 (0.23 U). Figure 5 shows the endo-chitinase enzyme activity of the transgenic potato plants.

The detached leaf assay was carried out on healthy leaves of SP-21 transgenic potato plants and non-control plants. In the Petri dish, an agar plug was placed on the transgenic and non-transgenic leaves on a moist paper towel. *Fusaruim oxysporum* had penetrated from the plug into the inoculation site of the leaves after 7 days. The pathogen was able to cause wilting on the control non-transgenic leaves after 7 days of inoculation, whereas pathogen spread on the transgenic leaves (SP-21) was delayed, as shown in Figure 6.

The relative mRNA expression of CP-PVX and PVY of the two positive transgenic lines (SP-21 and SP-148) for both viruses was quantified using qRT-PCR. The transgenic line SP-21 showed maximum knockdown mRNA expression compared to the control plant (non-transgenic infected plant with the virus). The knockdown expression of mRNA of CP-gene of both viruses was calculated according to [26]. As shown in Figure 7, the SP-21 transgenic line exhibiting shRNA for PVX and PVY showed 89% and 85% knockdown of mRNA against PVX and PVY, respectively, compared to the control. At the same time, the SP-148 transgenic lines showed 68% and 70% mRNA knockdown of PVX and PVY. Beta-actin was kept as an internal control for qRT-PCR. Morphologically, transgenic potato lines (SP-21 and SP-148) remained healthy, and there were no apparent symptoms of the virus compared to the non-transgenic control, which displayed mosaic and chlorosis after viral infection.

### 3.3. Viral Assays of Transgenic Plants

In total, 21 transgenic plants in the greenhouse harbouring the pCXY cassette were inoculated mechanically with PVY and PVX. ELISA was performed about 15 dpi, and 5 plants (SP-21, SP-57, SP-148, SP-249 and SP-290) were positive for PVY infection, while 7 plants (SP-21, SP-89; SP-148; SP-201; SP-260; SP-295 and SP-319) were positive for PVX infection.

## 4. Discussion

In modern agriculture, synthetic chemicals protect crop plants from various pathogens. However, excessive use of these chemicals is no longer viable due to bio-safety concerns such as environmental protection, public health, and increased pathogen resistance to fungicides. As a result, developing an effective and safe alternative for disease control in crop plants is required. In addition, fungi and viruses are pathogens of crops and cause diseases that affect the output of crops.

Potato, one of the world’s most important food crops, is susceptible to various fungal, bacterial, and viral pathogens [27]. It is widely infected by fungi throughout potato-producing regions, resulting in significant yield losses [28]. Fungi such as *Fusarium solani*, *Fusarium oxysporum*, *A. solani*, Pythium, *Rhizoctonia solani* and Phytophthora are the major pathogens of potatoes. *F.oxysporum* is responsible for vascular wilt disease in members of Solanaceae [29]. The cell wall of the fungi is composed of chitin and glucan. Chitin comprises N-acetyl glucosamine and is the second most abundant polysaccharide in the environment. The hydrolytic action of other enzymes, such as ß-1, 3-glucanase, activates the chitinase enzyme. Chitinase expression plays a vital role in controlling the disease; for example, high chitinase expression in roots might reduce or inhibit the growth of soil-born fungi and bacteria. When transformed in plants, plant chitinases have been shown to confer resistance to the fungal pathogen. The increase in the production of chitinases is shown to be associated with a decrease in pathogenic fungi and nematodes and, more importantly, with lessening infectivity and crop damage. Chitinases reduce the population of actinomycetes, fungi, and bacteria [19]. Chitinase was isolated from rice (*Oryza sativa*) and inhibited lesion development due to *Botrytis cinerea* [30]. Another study found that extracting chitinase from beans reduced the rate and total seedling mortality caused by *Rhizoctonia solani* [31]. Chitinase is also helpful in controlling tobacco diseases, such as *Botrytis cinerea*, *Rhizoctonia solani* and *Sclerotium rolfsii* [32]. Further, chitinase in tomatoes has also been identified to reduce plant disease incidence due to *Sclerotinia sclerotiorum* [33]. The expression of barley chitinase in potato (Desiree) inhibited *Alternaria solani*, the causative agent of early blight [22].

Worldwide, RNA viruses cause up to a 30% yield loss in potatoes, while the percentage rises to 80% in Pakistan [22]. PVX and PVY alone cause 30 to 50% yield loss, respectively, while in combination, the yield loss increases to 90% [4]. Potatoes are commonly propagated vegetatively; once infected, the virus spreads across successive generations, causing degeneration. As a result, secondary infections are more damaging than primary infections. RNAi technology can reduce target gene expression from any source, whether endogenous or viral. Most transgenic plants with virus resistance use an antisense sequence or a hairpin structure that forms dsRNA upon transcription. Double-stranded RNA (dsRNA) is essential in RNA silencing and is widely used to develop virus resistance [12]. Introduced transgenes that encode viral dsRNA enable the plant to recognize the invading virus subsequently. dsRNA produced from either a transgene or a replicating virus is cleaved into approximately 21 nucleotide fragments by the Dicer enzyme [34]. Virus-resistant transgenic plants that expressed artificial virus-specific hairpin RNAs (hpRNAs) or microRNAs (miRNAs) in host plants were developed using the RNAi mechanism [35,36].

This study aimed to develop transgenic plants resistant to fungi and viruses (PVX and PVY), for which the fungal-resistant chitinase gene and shRNA transgenes against the coat protein genes of PVX and PVY were ligated into the binary vector pCAMBIA2301 with CaMV35S promotor and NOS terminator. As a result, about 30 positive stable transgenic potato (AGB-R) lines were developed using Agrobacterium-mediated transformation. Furthermore, our study assessed the transformed potato variety AGB-R for antifungal activity against *Fusarium oxysporum*. The crude protein extracted from the control and transgenic plants was tested against *F. oxysporum*; the results in transgenic plants showed a maximum inhibition of ~65% in SP-21 and a minimum inhibition of ~14% in SP-89, as reported by [22], who evaluated chitinase activity against *A. solani* and found a maximum inhibition of 60.5 and a minimum inhibition of 39.5. Furthermore, a 40–56% inhibition of *C. falcatum* by crude protein in sugarcane transformed by using a barley chitinase II gene was reported by [37]; in tobacco plants, the inhibition of fungus was measured as 56.4–65.5% in transgenic lines [38].

In this study, an endochitinase activity assay was performed using cat# CS 0980 (Merck KGaA, Darmstadt, Germany). This assay indicates the amount of chitinase present in the crude protein of transformed plants required to inhibit p-nitrophenyl β-N, N′, N″ -triacetylchitotriose. A maximum activity of 1.19 U was seen in SP-21, and a minimum activity was observed in SP-89, measured as 0.23 U, with 5.67 U being the activity observed for the positive control provided in the kit. A similar assay was performed in a study by [39] gave a maximum endochitinase activity of 0.46 U mL^−1^, and [40] also reported a high value of endochitinase activity, whereas in Trichoderma strains, ranges between 0.014 and 0.051 U mL^−1^ [41] and 1.08 1.40 [42] were reported. Meanwhile, this range was 0.58–0.72 U mL^−1^ in sugarcane, where barley chitinase II was used to develop transgenic plants, as reported by [37].

The detached leaf assay also revealed increased resistance to the selected fungus (*F.oxysporum*) and overexpression of barley chitinase in transgenic plant leaves. After seven days of inoculation, the leaves of non-transgenic (control) plants displayed necrosis and marked chlorosis, whereas the leaves of transgenic plants bore minor or no symptoms. A similar study by [43] showed a marked difference in necrosis when leaves were infected with *R. solani*. A maximum of 41% necrotic lesions were observed in transformed plants, compared to leaves from non-transformed plants that exhibited a 90% necrotic area of lesions. In addition, 9 out of 16 were found to be highly resistant in the detached leaf assay, with a scale of the resistance value of more than 2 [44].

Potato plants were genetically modified to become virus-resistant by expressing shRNA, a hairpin-like RNA sequence that silences specific genes [45,46]. A thirty-day post-inoculation of PVX and PVY in transgenic and non-transgenic plants using qRT-PCR revealed that transgenic plants had deficient CP-PVY mRNA expression compared to non-transgenic potato plants. Only 2 out of the 21 plants inoculated with PVX and PVY tested positive for both viruses. The highest knockdown of mRNA for PVX and PVY was calculated as 89% and 85%, respectively, in SP-21, whereas it was 68% and 70% in SP-148, as compared to studies by [5] that showed an expression of 21.973% and [10] that reported more than 90% achievement in acquiring resistance against multiple viruses via the same approach. A study reported 100% resistance to PVY in potato lines using the same method [11]. Another study by [47] found that the expression of dsRNA from prokaryotes gave practical results of 71.68% against CP-PVY in tobacco plants. A tolerance of 40% and a resistance of 20% were reported by using siRNA in potato plants against CP of PLRV, PVX and PVY. Transgenic plants had mild yellowing symptoms and a mosaic appearance, whereas the control plants had severe symptoms and eventually died. Inoculation with a mix of PVX and PVY in transgenic plants demonstrated remarkable resistance, but non-transgenic plants exhibited enhanced symptoms, as observed in previous studies such as [20].

## 5. Conclusions

Potato, one of the world’s most important food crops, is susceptible to various fungal, bacterial, and viral pathogens The expression of chitinase and shRNA transgenes is an effective method for developing pathogen-resistant potato plants. Transgenic potato lines with anti-fungal and virus resistance were created by inoculating chitinase and shRNAs from the PVX and PVY CP genes. As a result, the developed potato cultivar AGB-R was resistant to fungal and viral pathogens.

## Figures and Tables

**Figure 1 genes-14-01090-f001:**
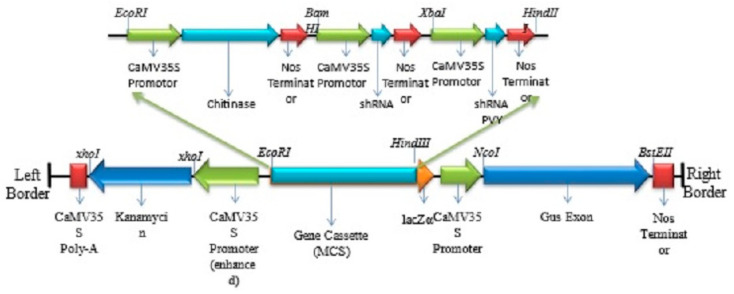
Construct details of chitinase and shRNAs of PVX and PVY in plant binary vector pCAMBIA2301. The cassette is cloned in the MCS region of the vector between EcoRI and HindIII restriction endonuclease.

**Figure 2 genes-14-01090-f002:**
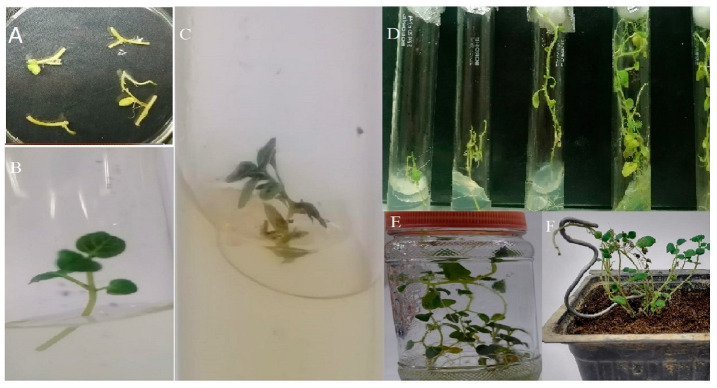
Steps in potato transformation and regeneration; (**A**) regeneration of transformed internodes in MS medium; (**B**–**D**) potato plantlets received after transformation in culture tubes; (**E**) jar; (**F**) acclimatized potato plant in a pot.

**Figure 3 genes-14-01090-f003:**
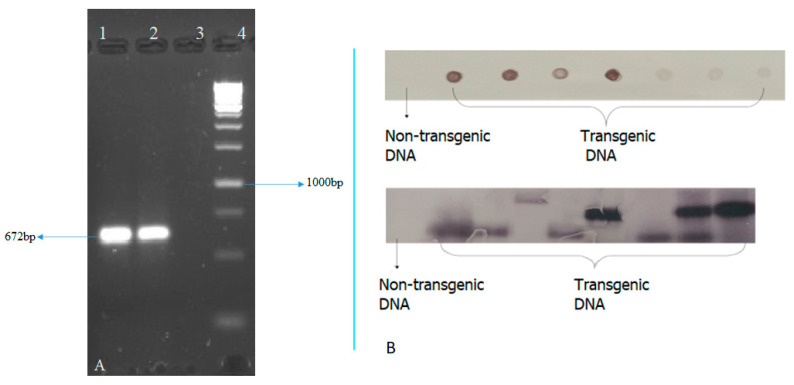
Molecular analysis of transformed potato plants; (**A**) PCR of chitinase gene present in the CXY construct, lanes 1–2 show positive plants, lane 3 shows negative non-transgenic control plant and lane 4 is a 1 kb ladder; (**B**) Southern blot of transformed potato plants.

**Figure 4 genes-14-01090-f004:**
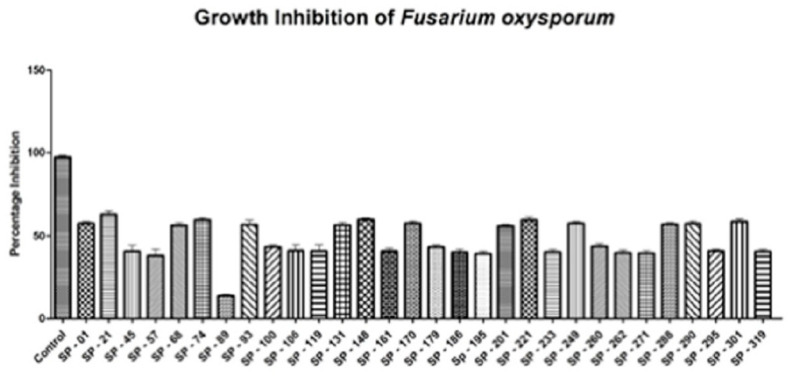
Growth inhibition of crude protein of transgenic potato plant against *Fusarium oxysporum*. X axis shows the individual transgenic plant, while the Y axis shows the percentage inhibition.

**Figure 5 genes-14-01090-f005:**
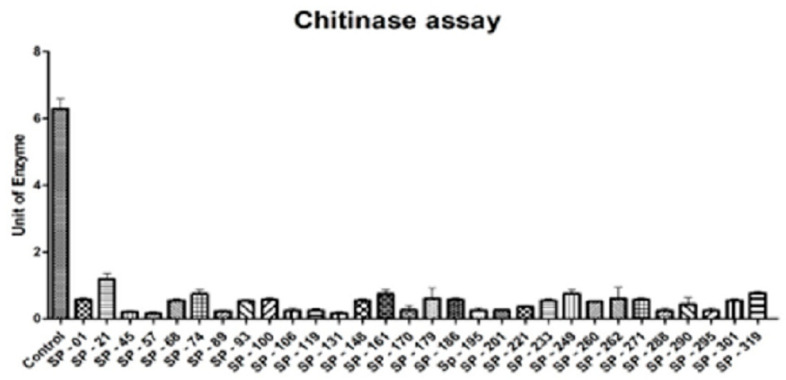
Chitinase activity assay of transgenic potato plants; X axis represents the transgenic potato plants used in this experiment, while the Y axis is the unit of enzyme required to hydrolyse the substrate.

**Figure 6 genes-14-01090-f006:**
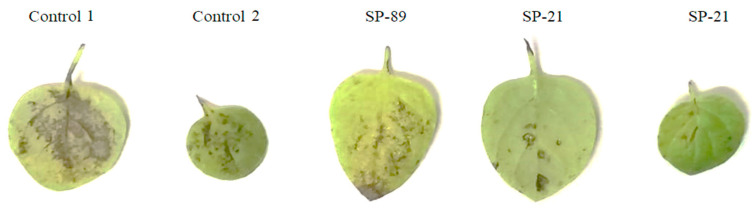
Detached leaf assay of transgenic potato plants with *Fusarium oxysporum* compared with the non-transgenic control.

**Figure 7 genes-14-01090-f007:**
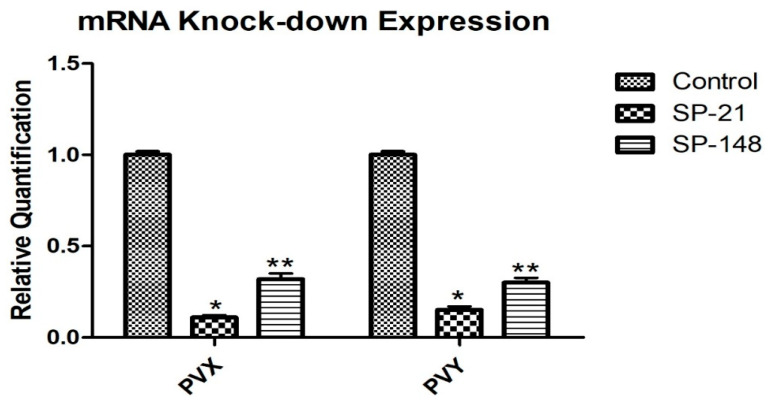
Relative quantification of mRNA knock-down expression of PVX and PVY in SP-21 and SP-148; the expression of CP mRNA is significantly lower in transgenic plants SP-21 and SP-148 when compared to the control (*p* < 0.05; *n* = 3; * and ** shows the significant value of mRNA knockdown compared to control).

**Table 1 genes-14-01090-t001:** The sequence of the primers used in this study along with their product size.

Name of Primers	Sequence of Primers	Amplicon Size
DCHIF	GTGATCACTCAATCAGTATACGC	672 bp
DCHIR	CCAAGCATACCGCAATACCT	
PVX-F	GGCAGCAGCAATTAAAGAGG	120 bp
PVX-R	GAAACCTTGTGCTTGCCAGT	
PVY-F	GCCAAATGTCAACGGAGTTT	100 bp
PVY-R	TTGCCTAAGGGTTGGTTTTG	

## Data Availability

The authors confirm that the data supporting the findings of this study are available within the article and the accession numbers of the genes used (if any) are available in Genbank.

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
