# Peer review of "Expression of Chitinase and shRNA Gene Exhibits Resistance to Fungi and Virus"

_genes, 2023, doi:10.3390/genes14051090_

Round 1

Reviewer 1 Report

To,

The Chief Editor,

Genes, MDPI,

Manuscript ID: genes-2294134

Subject: Submission of comments of the manuscript in “Genes"

Dear Chief Editor Genes, MDPI,

Thank you very much for the invitation to consider a potential reviewer for the manuscript (ID: genes-2294134). My comments responses are furnished below as per each reviewer’s comments. 

 Dear Chief Editor,

The reviewed manuscript authors developed potato lines resistant to fungi and virus, Potato Virus X (PVX) and Pota-to Virus Y (PVY), by inoculation of chitinase for fungi and shRNA designed against mRNA of Coat Protein of PVX and PVY respectively. The construct was developed us-ing the pCAMBIA2301 vector and transformed into AGB-R (red skin) potato cultivar using Agrobacterium tumefaciens. The crude protein extract of the transgenic potato plant inhibited the growth of Fusarium oxysporum from ~13 to 63%. The detached leaf assay of the transgenic line (SP-21) showed decreased necrotic spots compared to non-transgenic control when challenged with Fusarium oxysporum. The transgenic line, SP-21, showed maximum knockdown when challenged with PVX and PVY i.e. 89 and 86% while transgenic line SP-148 showed 68 and 70% knockdown in PVX and PVY respec-tive challenged conditions. It is concluded from this study that the developed trans-genic potato cultivar AGB-R showed resistance against fungi and viruses (PVX and PVY). The manuscript represents a very important piece of research in a logical presentation. Therefore, it might be conditionally accepted as subject to major revision. Instead, authors have to improve their manuscripts with many non-clear meanings, inaccuracies, and the authors need to address the following issues .

  1. I have read the entire manuscript and my initial comment is that manuscript is poorly written. I have significant concerns about the grammar and vocabulary of the manuscript; therefore, I recommend the authors to used an English proofreading service.. 
  2. The structure of the abstract should be improved, as well as the lack of several aspects that should be included in this section. Most of the abstracts contain confusing and uninformative sentences. Please give more precise objectives here (such as in the Abstract). The abstract should highlight the most important results of the parameters and characteristics assayed.
  3. Keyword must in alphabetical order.
  4. The figures are quite low resolution and difficult to make out. Higher-resolution versions will be needed for publication. Further, text in figure is not readble. for example, in Figures 1, 4, and 5.
  5. qRT-PCR methodology provided is also very vague and confusing. Please provide more details like what was the calibrator used in the study. I assume the authors have used the control as the calibrator. If so, the authors should not include the control within the bar graph as it represents the fold change between the treated vs control and a fold change of “1” for the ‘control’ doesn’t make any sense.  Also, would be good to provide details on what reagents (details of probes used, if any, if SYBR was used then details for that, etc.) and real time PCR machine were used in the current study.
  6. The discussion should be interpreted with the results as well as discussed in relation to the present literature.
  7. The conclusion section is very short. The author should emphasize this in a better way.
  8. References: shall have to correct the whole References according to the ”Instructions for the Authors”, e.g. the Journal name must be abbreviated, journal name in italics, the year must be bold and you shall have to use the abbreviated number of the Journals cited.

Author Response

Response to the comments of the Editor/Reviewer(s)

S.no.

Comments of Reviewer

Response to the comments

Position in Manuscript

1

I have read the entire manuscript and my initial comment is that the manuscript is poorly written. I have significant concerns about the grammar and vocabulary of the manuscript; therefore, I recommend the authors to use an English proofreading service

Extensive English editing done

Certificate attached

2

The structure of the abstract should be improved, as well as the lack of several aspects that should be included in this section. Most of the abstracts contain confusing and uninformative sentences. Please give more precise objectives here (such as in the Abstract). The abstract should highlight the most important results of the parameters and characteristics assayed.

Abstract improved as suggested by the reviewer

Line# 16-29

3

Keyword must in alphabetical order.

Changes made as suggested by the reviewer

Line # 30

4

The figures are quite low resolution and difficult to make out. Higher-resolution versions will be needed for publication. Further, text in figure is not readble. for example, in Figures 1, 4, and 5.

Changes made as suggested by the reviewer

5

qRT-PCR methodology provided is also very vague and confusing. Please provide more details like what the calibrator was used in the study. I assume the authors have used the control as the calibrator. If so, the authors should not include the control within the bar graph as it represents the fold change between the treated vs control and a fold change of “1” for the ‘control’ doesn’t make any sense.  Also, would be good to provide details on what reagents (details of probes used, if any, if SYBR was used then details for that, etc.) and real time PCR machine were used in the current study.

Relative expression of qPCR of transgenic potato lines were performed in challenged condition of potato virus X and Potato virus Y.

The non-transgenic control was kept as control and the mRNA expression of non-transgenic control infected with PVX and PVY was compared with transgenic potato lines SP-21 and SP-148.

Line# 316-319

Line# 413-422

6

The discussion should be interpreted with the results as well as discussed in relation to the present literature.

The whole discussion was arranged according to the authors guideline by the journal and as suggested by the reviewer.

7

The conclusion section is very short. The author should emphasize this in a better way.

Changes made as suggested by the reviewer

Line#637-642

8

References: shall have to correct the whole References according to the” Instructions for the Authors”, e.g. the Journal name must be abbreviated, journal name in italics, the year must be bold and you shall have to use the abbreviated number of the Journals cited.

Changes made as suggested by the reviewer

Reviewer 2 Report

The manuscript entitled “Expression of chitinase and shRNA gene exhibits resistance to fungi and virus” transgenic potato cultivar AGB-R was developed by using RNAi for potato virus X and Y along with the PR gene, chitinase, for fungi in plant binary vector pCAMBIA2301 via Agrobacterium-mediated transformation. The work is interesting but the confirmation at different level on transformants is not enough. Moreover, the regeneration and transformation pictures do not have good quality. Anyway, the manuscript needs a major revision. With the comments:

1- My main concern is about the confirmation of transformants. How many lines have been selected for these assays. Why there are only 2 plants for PCR confirmation and why the authors didn’t use uniform number of line for confirmation. For more confirmation you could do the PCR separately with primer pairs of each transgene.

2- However the manuscript is well-written, it needs a moderate English polishing for potential grammatical and structural errors throughout all parts of manuscript. As in some section, the sentences lack logical structure or lack relevant verbs and in some section the sentences are incomplete.

3- The quality of regeneration and transformation events is not satisfactory. To have a background please refer to the following references which has shown all the stages of regeneration and transformation:

 Vafaee, Y., & Alizadeh, H. (2018). Heterologous production of recombinant anti-HIV microbicide griffithsin in transgenic lettuce and tobacco lines. Plant Cell, Tissue and Organ Culture (PCTOC), 135, 85-97.

4- Technically southern blot should have marker to show the size of fragments. Furthermore, in most cases more than 1 transgene integrate into the genome. It seems that southern blot image has been cropped please bring the whole image.

5- Why the fungal test has only tested on detached leaf not at whole plant scale?

6- To see the position effect, more transgenic line should be analyzed particularly at DNA and RNA scales.

7- There is no information in material and method how the viruses have been obtained an how plant inoculated.

8- The amplified fragment shown in Fig 3 is just show the chitinases gene. How you confirmed the integration of shRNA’s of PVX, and PVY in potato genome?

9- Please provide all information for reagent and kit providers (company and country name).

10- Please insert subtitle number throughout of the manuscript.

11- Why the authors didn’t purify the enzyme and its revealing on SDS-PAGE?

12- Last but not least, is there any overlapping between this work with the previous published work:

Khan, A., Nasir, I. A., Tabassum, B., Aaliya, K., Tariq, M., & Rao, A. Q. (2017). Expression studies of chitinase gene in transgenic potato against Alternaria solani. Plant Cell, Tissue and Organ Culture (PCTOC), 128, 563-576.

Author Response

Reviewer 2

S.no.

Comments of Reviewer

Response to the comments

Position in Manuscript

1

My main concern is about the confirmation of transformants. How many lines have been selected for these assays. Why there are only 2 plants for PCR confirmation and why the authors didn’t use uniform number of lines for confirmation. For more confirmation you could do the PCR separately with primer pairs of each transgene.

In total 162 regenerated plants were assessed for transgene with the concept that if one transgene is positive these mean plants are positive for the complete construct.

The authors can provide the gel pic for all three transgenes.

The two plants for PCR is symbolic in actual 162 plants were assessed and 30 plants were positive.

2

However, the manuscript is well-written, it needs a moderate English polishing for potential grammatical and structural errors throughout all parts of manuscript. As in some section, the sentences lack logical structure or lack relevant verbs and in some section the sentences are incomplete.

Extensive English editing done

Certificate attached

3

the quality of regeneration and transformation events is not satisfactory. To have a background please refer to the following references which has shown all the stages of regeneration and transformation:

 Vafaee, Y., & Alizadeh, H. (2018). Heterologous production of recombinant anti-HIV microbicide griffithsin in transgenic lettuce and tobacco lines. Plant Cell, Tissue and Organ Culture (PCTOC), 135, 85-97.

Figure 2 modified as suggested by the reviewer.

4

Technically southern blot should have marker to show the size of fragments. Furthermore, in most cases more than 1 transgene integrates into the genome. It seems that southern blot image has been cropped please bring the whole image.

In this manuscript we have not done the copy number southern blot. The authors have done the southern blot for the integration of the transgene into the plant.

5

Why has the fungal test only tested on detached leaf not at whole plant scale?

The fugal test was performed using.

a) Crude extract of transgenic line

b) Chitinase assay

c) Detach leaf assay

The whole plant scale was performed but the authors are planning to submit the pot trial and the field trial results in separate manuscript.

If the reviewer still feels that the whole plant data is necessary, the authors will include it in the manuscript.

6

To see the position effect, more transgenic line should be analyzed particularly at DNA and RNA scales.

In total of 320 nodes were used for the transformation of the construct and only 162 nodes regenerated and only 30 plants were PCR positive.

Line# 216-226

7

There is no information in material and method how the viruses have been obtained an how plant inoculated.

The viruses were provided by Dr. Bushra Tabassum, School of biological sciences, University of the Punjab.

The virus infected leaves were grounded using phosphate saline solution and the sap of the grounded leaves were used to inoculate further plants.

Line# 167-168

Line# 173-175

Line# 183-186

8

The amplified fragment shown in Fig 3 is just show the chitinases gene. How you confirmed the integration of shRNA’s of PVX, and PVY in potato genome?

The shRNAs were confirmed after ligation into the vector, pCAMBIA2301, using restriction enzymes BamHI and HindIII.

Line#89-91

9

Please provide all information for reagent and kit providers (company and country name)

Changes made as per suggested by the reviewer

Line#205; Line#228

Line#244; Line#277

10

Please insert subtitle number throughout of the manuscript.

Changes made as per suggested by the reviewer

All over the manuscript

11

Why didn’t the authors purify the enzyme and its revealing on SDS-PAGE?

SDS-PAGE for the chitinase was already purified and published by the author Khan, A., Nasir, I. A., Tabassum, B., Aaliya, K., Tariq, M., & Rao, A. Q. (2017). Expression studies of chitinase gene in transgenic potato against Alternaria solani. Plant Cell, Tissue and Organ Culture (PCTOC), 128, 563-576). In this article the authors tried to combine all three genes and develop potato lines tolerant to fungi and virus.

12

Finally, is there any overlapping between this work with the previous published work:

Khan, A., Nasir, I. A., Tabassum, B., Aaliya, K., Tariq, M., & Rao, A. Q. (2017). Expression studies of chitinase gene in transgenic potato against Alternaria solani. Plant Cell, Tissue, and Organ Culture (PCTOC), 128, 563-576.

In the previous work (Khan, A., Nasir, I. A., Tabassum, B., Aaliya, K., Tariq, M., & Rao, A. Q. (2017). Expression studies of chitinase gene in transgenic potato against Alternaria solani. Plant Cell, Tissue and Organ Culture (PCTOC), 128, 563-576) the authors focus on developing antifungal potato line, desiree, only. Based on the finding of the previous research the authors combined three gene to develop antifungal and antivirus potato lines, AGB-R.

Round 2

Reviewer 1 Report

Dear Editor,

Thank you for providing the opportunity to review the revised manuscript. The manuscript is improved considerably after revision according to the reviewer's comment. Now this study is a suitable contribution to the Gene. I recommend the manuscript for publication.

Thank you

With best regards

Author Response

S.no.

Comments of Reviewer

Response to the comments

Position in Manuscript

1

In my opinion, the respected authors have already done the required revisions, however, for more clarification, the virus test on the whole plant should also be added. So, the manuscript now needs minor revision.

The virus (PVX and PVY) was challenged on quarantined plants present in the green house.

The data of real time (relative qPCR) expression of CP-PVX and PVY of transgenic and non-transgenic control plant presented in figure 7 is of greenhouse quarantined potato plants

Line# 288-295

Line# 304-311

Line# 426-444

Reviewer 2 Report

In my opinion, the respected authors have already done the required revisions, however, for more clarification, the virus test on the whole plant should also be added. So the manuscript now needs minor revision.

Author Response

Dear Sir, 

S.no.

Comments of Reviewer

Response to the comments

Position in Manuscript

1

In my opinion, the respected authors have already done the required revisions, however, for more clarification, the virus test on the whole plant should also be added. So, the manuscript now needs minor revision.

The virus (PVX and PVY) was challenged on quarantined plants present in the green house.

The data of real time (relative qPCR) expression of CP-PVX and PVY of transgenic and non-transgenic control plant presented in figure 7 is of greenhouse quarantined potato plants

Line# 288-295

Line# 304-311

Line# 426-444

Reviewer 2
